# A Robust Deep Feature Extraction Method for Human Activity Recognition Using a Wavelet Based Spectral Visualisation Technique

**DOI:** 10.3390/s24134343

**Published:** 2024-07-04

**Authors:** Nadeem Ahmed, Md Obaydullah Al Numan, Raihan Kabir, Md Rashedul Islam, Yutaka Watanobe

**Affiliations:** 1Department of Computer Science and Engineering, University of Asia Pacific, Dhaka 1216, Bangladesh; nadeem@uap-bd.edu; 2Department of Computer Science and Engineering, University of Aizu, Aizu-Wakamatsu 965-8580, Japan; obaydullahhasib.cse@gmail.com (M.O.A.N.); raihan.kabir.cse@gmail.com (R.K.); yutaka@u-aizu.ac.jp (Y.W.)

**Keywords:** ambient assisted living, human activity recognition, IMU, inertial sensors, wearable sensors, time-frequency analysis, wavelet transform, continuous wavelet transform, scalogram, deep learning, classifier

## Abstract

Human Activity Recognition (HAR), alongside Ambient Assisted Living (AAL), are integral components of smart homes, sports, surveillance, and investigation activities. To recognize daily activities, researchers are focusing on lightweight, cost-effective, wearable sensor-based technologies as traditional vision-based technologies lack elderly privacy, a fundamental right of every human. However, it is challenging to extract potential features from 1D multi-sensor data. Thus, this research focuses on extracting distinguishable patterns and deep features from spectral images by time-frequency-domain analysis of 1D multi-sensor data. Wearable sensor data, particularly accelerator and gyroscope data, act as input signals of different daily activities, and provide potential information using time-frequency analysis. This potential time series information is mapped into spectral images through a process called use of ’scalograms’, derived from the continuous wavelet transform. The deep activity features are extracted from the activity image using deep learning models such as CNN, MobileNetV3, ResNet, and GoogleNet and subsequently classified using a conventional classifier. To validate the proposed model, SisFall and PAMAP2 benchmark datasets are used. Based on the experimental results, this proposed model shows the optimal performance for activity recognition obtaining an accuracy of 98.4% for SisFall and 98.1% for PAMAP2, using Morlet as the mother wavelet with ResNet-101 and a softmax classifier, and outperforms state-of-the-art algorithms.

## 1. Introduction

The use of human activity recognition (HAR) is expanding rapidly. To identify regular human activity, abrupt changes in the activity in real-time can provide important information in relation to the targeted concerns. Thus, HAR techniques are playing a significant role in elderly care, investigation activities, healthcare, sports, smart homes, surveillance activities, context-aware computing, athletics, and so on [1]. In recent research, artificial intelligence and robots are used to perceive the world around them, and to interact with humans and the environment. Therefore, human activity recognition (HAR) has become an integral part of robotics applications that aim to interact with humans. HAR is often regarded as a challenging task as humans perform numerous stationary and non-stationary activities in different ways where the activities are often organized into several sub-activities, i.e., defining complex activities [2]. HAR data can be generally gathered in two ways, namely, sensor-based and image-video-based, where each has its own merits and demerits.

While the location of an item ensures that a camera can continuously collect accurate data, in many complex scenarios, data collection may be influenced by motion blur, angle, objects in the way, and variations in illumination [3]. Activity classification involves image processing technologies. However, there are significant problems with utilizing an image to distinguish distinct activities as it can infringe on consumer privacy. This is a major concern in several applications. For instance, it is not enough to merely set up cameras to keep a watch on the actions of senior adults living in care facilities. That will lead to major privacy difficulties. Researchers support the use of sensors over other methods because of their several advantages [4]. They are lightweight, easily portable, mountable at various locations, have low power consumption, have comparatively low energy consumption, and are fully private. Since the sensor placements are not fixed [5], the sensors may produce changing data. We have consequently concluded that sensor data may be misleading in the absence of a very efficient classifier. Sensor-based technology’s major goal is to implant wearable electronics within the human body. Sensors on wearable technology can generate data depending on the direction of movement of the user. An accelerometer and gyroscope, which are widely employed for activity identification, make up the majority of inertial sensors. Their integration is visible in wearable technology, including fitness trackers, Fitbits, smartphones, smartwatches, and more [6].

Data from sensors are produced in the form of a time series, which is chronological. Many researchers have used time-domain, particularly frequency-domain, analysis to extract the intrinsic information for activity recognition. Useful information can generate relevant features that can be fed to the machine and deep learning model to recognize the activity [7,8,9,10,11]. Research outcomes indicate that recognition accuracy is quite good. However, in time-domain analysis, only fine resolution in time is available with no frequency information provided at that moment.

The opposite is also true in the case of frequency-domain analysis [12]. Thus, the information obtained is always lagging behind both when extracting the relevant statistics for activity recognition. Alternatively, time-frequency-domain analysis offers a much better combination of time- and frequency- domain features from the signal supporting optimal performance in many application. Some widely used representative time-frequency-domain methods include short-time Fourier transform (STFT), the Wingner–Vile distribution, and the wavelet transform method. The STFT shows good time resolution but poor frequency resolution [13]. The Wigner–Ville distribution is a time-frequency representation that provides the best possible time and frequency resolution [14]. However, it is also very sensitive to noise and can produce interference terms [15]. The wavelet transform is a more sophisticated time-frequency representation, which uses a set of basis functions called wavelets to decompose the signal into different frequency bands. Wavelets have good time and frequency resolution [16]. This allows for a more flexible and adaptive representation that captures both time and frequency information effectively. Wavelets possess the advantage of providing good time and frequency resolution simultaneously. The key characteristic that sets the wavelet transform apart is its ability to localize in both domains allowing for a more precise analysis of signal components with varying frequencies at different time intervals. It helps to analyze and process the nonlinearities in the signals that are often encountered in the real world. The purpose of transforming signal data into image representations, such as spectrograms or scalograms, instead of relying on real images, stems from the significant advantages offered within the domain of deep learning models for activity recognition, primarily due to the model’s efficiency in feature extraction. This approach proves particularly advantageous given the inherently temporal nature of sensor signals. Converting these sequential signals into visual representations in the form of images captures both temporal and frequency information effectively. This streamlined representation facilitates a more efficient process of feature extraction by deep learning models, enabling them to discern intricate temporal patterns and variations in human activities more proficiently. A noteworthy reference supporting this approach is the work of Ronao et al. [17], where the paper explores the widespread use of time-frequency representations for translating sensor signals into images. This underscores the efficacy of this method in capturing nuanced temporal patterns crucial for accurate activity recognition. In essence, utilizing image representations aligns seamlessly with the strengths of deep learning, optimizing the model’s ability to extract meaningful features from the temporal dynamics of sensor data. Various methods have been devised, including the wavelet kurtogram, spectrogram, scalogram, and several other techniques rooted in the principles of wavelet transform [18]. The most commonly used technique in time-frequency analysis is the wavelet kurtogram. It is derived by applying the wavelet transform to the signal and then computing the kurtosis at each scale to reveal the non-Gaussian behavior at different scales, helping identify regions in the signal. It can be used for HAR, but Wavelet kurtograms are especially effective in detecting sharp changes in the signal; it is not possible to achieve a good result for detecting HAR due to its sensitivity to quick shifts in the signal, which might not always mean someone is performing a specific activity. Alternative methodologies have been developed to overcome this limitation. For instance, the spectrogram [19] is especially valuable when analyzing signals with consistent features, offering a visual representation of a signal’s spectral content over time. It excels in the examination of stationary signals, where stable frequency components enable a clear depiction of spectral variations. However, HAR signals are typically non-stationary, involving dynamic and changing patterns. In such cases, the scalogram [20] method becomes particularly useful for HAR applications. Its strength lies in its adaptability to non-stationary signals, providing a visual representation of time-varying frequency content in a manner that complements the dynamic nature of activities in HAR scenarios and is particularly useful for analyzing signals with non-stationary characteristics. On the other hand, recent research indicates that deep learning-based models exhibit exceptional performance in the realm of 2D space-image analysis, demonstrating a robust capability to classify various activities [21]. Leveraging this insight, scalograms emerge as a highly effective means to formulate the required spectral images [22], comprehensively representing a signal’s time-varying frequency content. This information-rich representation aligns seamlessly with the strengths of deep learning methodologies, which excel in extracting intricate patterns from complex data. HAR is essential for smart technologies such as elderly care, surveillance, and personalized health monitoring. Cameras and sensors are two robust procedures for identifying the different activities. Relying solely on cameras for activity recognition poses challenges and generates privacy concerns. Researchers prefer sensor-based technologies for their portability, low power consumption, and privacy preservation. Wearable devices with accelerometers and gyroscopes provide valuable data for activity identification. Traditional time- and frequency-domain analyses of sensor data lack simultaneous information, prompting the adoption of time-frequency-domain methods, like wavelet transforms. However, for efficient feature extraction in deep learning models, transforming sensor data into image representations, such as scalograms, is favored. This approach optimizes the model’s ability to discern intricate temporal patterns. Scalograms, which are particularly effective in HAR scenarios, leverage deep learning’s strengths in analyzing 2D space-image data for accurate activity classification. Keeping this in mind, in this paper, firstly, the 1D data have been formulated and converted to scalogram images by time-frequency-domain analysis and the features extracted by deep learning; then, a conventional classifier is used to identify the human activity for use in the elderly care system. In summary, the key contributions of this research include:This paper proposes a model that converts activity signals into scalogram images and processes the images into a 3-channel representation. Subsequently, the deep features are extracted from activity images generated from the scalogram using deep learning models.An exploration of the optimal combination of mother wavelets for scalogram generation, coupled with deep learning models, was undertaken to enhance accuracy and efficacy in human activity detection.Additionally, a comprehensive experiment was conducted using two renowned datasets focusing on HAR, encompassing a variety of daily activities. This experiment served to validate the proposed model in terms of its efficiency, robustness, and scalability.

In this paper, the model is described and a detailed literature review is provided in Section 2, delving into sensor-based data preferences and advanced techniques like the wavelet and continuous wavelet transform. The proposed model is then introduced in Section 3, leveraging convolutional neural networks (CNNs) such as ResNet, GoogLeNetV3, and AlexNet, emphasizing efficient learning through softmax. The experimental setup and analysis of results are presented in Section 4, covering dataset selection, preprocessing, and hyper-parameter tuning. The model performance metrics are interpreted, highlighting strengths and areas for improvement. In Section 5, the model is benchmarked against the existing literature, showcasing advancements. The paper concludes in Section 6 by synthesizing key findings and discussing future research directions, offering a holistic view of the contributions to HAR.

## 2. Literature Review

To date, numerous types of research have been conducted on activity recognition. Broadly, activity recognition can be divided into two parts: vision-based and sensor-based. Many researchers have concentrated on images and video, whereas other researchers have concentrated on sensor-based signals. However, most of them are activity class identification tasks where different artificial learning models, such as the Hidden Markov Model, machine learning, and deep learning approaches, are used for activity classification. Below, we first discuss some important research about visual media, followed by discussion of sensor-based signals.

### 2.1. Vision-Based HAR

In recent investigations into human activity recognition, researchers have explored diverse methodologies to enhance accuracy and address specific challenges faced by vision-based HAR. In a recent study by Hao et al., the researchers observed that color-depth local spatio-temporal features (CoDe4D) are generated from RGB-D videos. They evaluated the performance of CoDe4D features combined with the bag-of-features (BoF) encoding representation. The traditional SVM classifier was used for activity recognition. They applied their method in four datasets, namely, UTK Action3-D, Berkeley MHAD, and ACT4, where the activities were different from each other [23]. However, in many cases, a convolutional neural network (CNN) was used for activity classification. Koki et al. used ensembled transfer learning to recognize activity from a motionless image. To build feature fusion-based ensembling, four CNN branches were applied, and in each branch, an attention module was employed to gather contextual data from the feature map established by previously trained models. The final recognition output was constructed on three datasets: Stanford 40 actions, BU-101, and Willow-Actions. The recovered feature maps were concatenated and provided to a fully connected network [24]. Hazar et al. used a CNN model on video sequences of the UCF-ARG dataset. This worked on an offline phase and an inference phase where a scene stabilization step was performed. Classification happened in two phases: identifying the video frames and identifying the whole video arrangements [25]. In Disha Deotale et al., the authors identified some gaps such as limited accuracy, scalability, and applicability as the classification of a large number of frames creates large delays. To minimize the weakness, a sub-activity stitching model (GSAS) was proposed based on an innovative gated recurrent unit (GRU). It worked on two stages: GRU, a sub-activity identification, and a 1D convolutional neural network (CNN), a sub-activity edging on the untrimmed datasets [26]. Subsequently, Yujia Zhang et al. identified that traditional data augmentation is crucial for human activity classification. According to their study, traditional approaches like data cropping are generally responsible for bad sample generation. For this reason, overall performance is hampered. The authors thus proposed a non-complex operative method known as Siamese architecture and labeled it as a Motion-patch-based Siamese Convolutional Neural Network (MSCNN). They evaluated and experimented with well-known datasets such as UCF-101 and HMDB-51 [27]. Alongside machine and deep learning, Ivan et al. used a statistical Hidden Markov Model for the activity recognition of human activity. A body poses dictionary was used to obtain the spatial and temporal compositions of atomic human actions. Two complex activity datasets, the MSR-Action3D and the Composable Activities dataset, were used in this research [28].

### 2.2. Sensor-Based HAR

These days, wearable sensors, artificial neural networks, and HAR technologies dramatically influence the quality of life every day. These technologies have attracted the attention of several organizations and researchers who wish to improve all elements of human life. Owing to sensor-based HAR’s increased privacy features and exponential growth, various effective supervised machine learning algorithms have been applied in studies. Sensor-based HAR makes use of a range of sensors, including Bluetooth, accelerometers, gyroscopes, and sound sensors.

Three inertial sensor units on upper and lower body limbs were employed by Attal et al. to research wearable sensor-based human activity recognition. Using the RF method as a wrapper for feature selection, they contrasted supervised (k-NN, SVM, GMM, RF) and unsupervised (k-Means, GMM, HMM) classification approaches [29]. The results showed that when it comes to unsupervised categorization in the context of everyday activities utilizing three wearable accelerometers, HMM is a a superior alternative to k-NN for supervised techniques. Many supervised machine learning approaches have been employed in HAR with significant results [1]. Among these, deep learning stands out as being exceptionally effective and exact in pattern recognition, eliminating the need for manual feature construction. Because it can automatically learn relevant and high-level properties through end-to-end neural network training, it is the best option for HAR applications. In this research paper [30], a deep recurrent neural network (DRNN) is employed to provide a framework for high-throughput human activity recognition from raw accelerometer data. Some recent papers show that converting sensor data into images before training with deep learning can lead to better performance on a variety of tasks, including footstep detection [31], human activity recognition [32], and fall detection [33]. The main reason for this improvement is that converting sensor data into images allows the deep learning model to learn spatial features from the data. Spatial features are important for many tasks, such as identifying objects and recognizing activities. Jaegyun Park et al. [34] realized the importance of sensor-based human activity recognition using time series signals. The authors identified that traditional systems sought to use non-concrete pure signals by sampling with a predefined interval, making it difficult to gain the desired accuracy as the real sample is previously unknown. Thus, the authors proposed a novel multi-temporal sampling module that uses multiple sampling intervals instantaneously in the neural network on the PAMAP2 dataset [28].

Saeedeh Zebhi used two domain analysis techniques, WVT and 2D FFT (two-dimensional fast Fourier transforms). After sensor data were displayed onto an image display, CNN was used to classify activities. The UCI HAR, MOTIONSENSE, MHEALTH, and WISDM datasets were used in this investigation [35]. Sakorn Mekruksavanich et al. used sensor data to tackle complex HAR problems using a deep neural network consisting of convolutional layers and residual networks with a squeeze-and-excite technique. The effectiveness of the model was tested using three publicly available datasets (WISDM-HARB, UT-Smoke, and UT-Complex) [36]. Wassila Dib presented a HAR study that measured the Received Signal Strength Indicator (RSSI) of several body channels on-body using a threshold-based methodology. To identify both static and dynamic behaviors, the authors used typical machine learning techniques combined with unconventional statistical characterization factors [37]. Imen et al. introduced CWT-CNN2D, a novel approach integrating Continuous Wavelet Transform (CWT) with 2D Convolutional Neural Networks (CNNs) for recognizing human activities using IMU data from smartphones and wearables across various datasets. CWT converts accelerometer and gyroscope signals into image representations, which are then fed into a 2D CNN for classification. Compared to baseline methods, such as RF and CNN-LSTM, CWT-CNN2D achieves impressive accuracy of 93.9% on the UCI HAR dataset. However, its performance on the Pamap2 dataset is notably lower, achieving only 74.3% accuracy [38]. The study by Nadia et al. presents a CNN-MLP hybrid model for sensor-based human activity recognition, leveraging deep learning’s autonomous data insights. Achieving 97.14% accuracy on the UCI HAR dataset, it integrates CNN and MLP layers for effective feature extraction and pattern recognition [39]. Yasin et al. conducted a study on sensor-based human activity recognition using deep 1D-CNNs, evaluating their approach on two datasets, UCI-HAPT and PAMAP2, achieving accuracy rates of 98% and 90.27%, respectively. Their model leveraged raw accelerometer and gyroscope data, with parameter optimization for refining architecture and hyper-parameters. They highlighted the enhanced performance gained from integrating sensor data compared to using individual sensors alone [40]. Zebin et al. introduced a deep learning model for sensor-based human activity recognition (S-HAR), leveraging both inertial and stretch sensors. Tested on the w-HAR dataset, their model achieved high accuracy (97.68%), outperforming existing methods. This study highlighted the effectiveness of combining sensor data types and demonstrated strong generalization in activity classification [41].

## 3. Proposed Methodology

The proposed HAR methodologies consist of three main processes, including data acquisition, time-frequency-domain analysis, and activity classification. Figure 1 illustrates the overall flowchart of the proposed methodology. The first part is the data collection process from a wearable sensor. The second part of the analysis involves time-frequency-domain analysis since the dataset comprises time-domain signal data. We adopt a wavelet-based scalogram data analysis model for generating spectral images, enabling the identification of relevant features in the time-frequency domain. The third part consists of activity detection to identify the regular activity using a deep learning model.

### 3.1. Sensor Data Acquisition

The research employs wearable sensors, particularly accelerometers and gyroscopes for generating activity data. The sensor data utilized in this study are sourced from two well-known publicly available datasets. Both the accelerometer and gyroscope sensors generate signals in three axes (X, Y, Z) for each activity recorded in the datasets. The accelerometer collects data from the body, measuring specific forces in three dimensions, while the gyroscope records rotational signals across the same axes. An IMU (Inertial Measurement Unit) integrates these sensors to report on a body’s specific force, linear acceleration, angular rate, and orientation, encompassing various daily activities. This paper focuses on daily activities characterized by linear acceleration, tilting, and vibration. In the analysis of daily life activities, features related to body twisting and rotational movements are deemed less critical. Therefore, accelerometer data were prioritized for feature extraction to enhance the efficiency of daily activity recognition. The accelerometer’s precise capture of linear acceleration aligns better with targeted movements, such as walking and running. Consequently, a more streamlined data processing pipeline is achieved and power consumption is reduced. The decision to utilize accelerometer data in generating scalogram images stems from a deliberate evaluation process. Gyroscope data encompass information about rotational movements and alterations in orientation. However, thorough analysis of related work revealed that these additional sensor inputs did not significantly enhance the recognition of the specific set of activities under consideration [42,43]. Figure 2 illustrates the variations in signals for various basic activities by showing three axes of the accelerometer sensor, including walking (Figure 2a), jumping (Figure 2b), running (Figure 2c), and standing (Figure 2d).

### 3.2. Time-Frequency Analysis and Scalogram Generation

Time-frequency analysis is a technique used to analyze signals whose frequency content varies over time. It provides a way to understand how the spectral characteristics of a signal change over time, which is crucial for non-stationary signals. A scalogram is a visual representation of this analysis, created by plotting the squared magnitude of wavelet coefficients in a time-frequency plane. This representation highlights the signal’s frequency components as they evolve, enabling detailed examination of complex signals in various fields, such as HAR, engineering, medicine, and finance. Using time-frequency-domain values rather than raw data for HAR offers significant advantages in feature extraction and classification performance. Raw sensor data, consisting of time series data, often fails to capture frequency-related information essential for distinguishing activities. Time-frequency analysis, such as use of wavelet transforms, enables simultaneous examination of time and frequency characteristics, providing a comprehensive signal representation. This dual-domain analysis reveals detailed patterns and changes within the signal, enhancing the recognition of complex activities. The Fourier Transform (FT) represents a signal in the frequency domain but loses information about the time at which certain frequencies occur. This limitation becomes evident in cases where the systems change their physical properties and characteristic spectrum over time. While the FT spectrum is easily interpreted for stationary systems, it fails to directly correlate temporal signal modifications with frequency features. For non-stationary signals, where the physical properties and characteristic spectrum of the signal change over time, the FT alone fails to directly correlate the temporal signal modifications with the frequency features of the spectrum. The FT represents the spectrum integrated over the acquisition time, making it challenging to capture the signal’s time-varying behavior. To overcome this limitation, methods that combine time- and frequency-domain analysis are needed to show the signal’s evolution in both domains. Windowed FT gives both spectral and temporal resolution, and it was among the first algorithms to work in the time-frequency plane. However, because the size of the time-frequency window is controlled by the window function employed, the windowed FT has a fixed time-frequency resolution. A superior option, however, is supplied by the wavelet transform (WT), which gives a time-frequency representation of a signal with variable time-frequency resolution.

#### 3.2.1. Continuous Wavelet Transform (CWT)

This paper uses Continuous Wavelet Transformation (CWT) to extract valuable time-frequency information from the signal data representing various human activities. It is a powerful tool for time-frequency analysis, particularly suited for capturing the nuanced dynamics of nonstationary signals where frequency content fluctuates over time. Unlike the Discrete Wavelet Transform (DWT), which has limited support in time [44], the CWT excels in providing a detailed portrayal of signal features across multiple scales and frequency bands [45]. Leveraging its ability to efficiently analyze signal breakdown into numerous frequency bands, the CWT proves valuable in identifying subtle variations, trends, and shifts in human activity. Human activities have quickly varying characteristics and CWT is superior in time-frequency resolution, making CWT the preferred choice in this paper.

CWT is a method of representation of the real-valued function 
S(t)
 as the following integral (Equation 1):
(1)
$$W_{s}\left(a,b\right)=\frac{1}{\sqrt{a}}\int_{\infty }^{-\infty }S\left(t\right)\Psi \left(\frac{t-b}{a}\right)dt$$


Based on a scale 
$a>0(a\epsilon R^{+})$
 and translocational value 
b(bϵR)
. The Discrete Wavelet Transform (DWT) carries with it a similar principle [16], with the difference that the parameters *a* and *b* are discrete:
(2)
$$a={\left(a_{0}\right)}^{n},\;b=kb_{0}$$


As was previously noted, CWT offers a great way to extract and study complex spectral information from a signal. The function is referred to as a mother wavelet since it is continuous in both time and frequency. For every conceivable pair 
(a,b)
, this mother wavelet is utilized to create a daughter wavelet:
(3)
$$\Psi_{a,b}\left(t\right)=\frac{1}{\sqrt{a}}\Psi \left(\frac{t-b}{a}\right)$$


Following that, the CWT is implemented:
(4)
$$\begin{array}{cc} W_{s}\left(a,b\right) & =\frac{1}{\sqrt{a}}\int_{-\infty }^{\infty }S\left(t\right)\Psi \left(\frac{t-b}{a}\right)dt\\ \; & =\int_{-\infty }^{\infty }S\left(t\right)\Psi_{a,b}\left(t\right)\;dt \end{array}$$


The similarity between the signal in question and each of the offspring wavelets is indicated by the Formula (Equation 4). An image with the b-value set on the x-axis and the a-value set on the y-axis can be used to illustrate these results. Mother wavelet functions require three extra things as follows: First, this function has to be restricted. Its squared module (Equation 5) must, therefore, be constrained.

(5)
$${\left||\Psi |\right|}_{2}=\int_{-\infty }^{\infty }{\left||\Psi \left(t\right)|\right|}^{2}dt<\infty$$


Second, the function has to be localized both in time and in frequency. Finally, the area under the curve has to be zero. Selecting an appropriate mother wavelet is crucial for effective feature extraction, considering the unique properties of different wavelets tailored to specific features. The choice is guided by the need to capture signal characteristics, with scales determining the frequency range for global or localized focus. The convolving process at each scale yields coefficients representing alignment, enhancing the overall feature extraction.

Mother Wavelet: The success of the CWT relies on the choice of an appropriate mother wavelet function 
Ψ(t)
, a mathematical function that is used to generate other wavelets through scaling and translation. Wavelets are small, localized functions that can be used to represent the different frequency components of a signal at different scales. Mother wavelets are typically chosen to have specific desirable properties, such as vanishing moments, which makes them well-suited for denoising applications, or orthogonality, which makes them easy to compute. There are several widely used mother wavelets, but in this paper, four of them—Morlet [46], Mexican Hat [47], Complex Morlet [48], and Shannon [49]—are applied. The Morlet wavelet function is defined as (Equation 6):
(6)
$$\psi \left(t\right)=A\cdot e^{-\frac{t^{2}}{2\sigma^{2}}}\cdot e^{i2\pi f_{0}t}$$


The Morlet wavelet as a function of time *t* is represented by 
ψ(t)
 in this equation. The wavelet has unit energy because of the term *A*, which serves as a normalizing constant. The wavelet is temporally localized via the Gaussian envelope 
$e^{-\frac{t^{2}}{2\sigma^{2}}}$
, whose width is set by 
σ
. A complex exponential term is created by adding 
$e^{i2\pi f_{0}t}$
, which produces an oscillatory component with a frequency of 
$f_{0}$
. The Morlet wavelet’s intrinsic complexity allows it to efficiently extract both the frequency and the temporal information from a signal. The Complex Morlet wavelet is defined as follows (Equation 7):
(7)
$$\psi \left(x\right)=\frac{1}{\sqrt{\pi \cdot fb}}\cdot \exp\left(2\pi i\cdot fc\cdot x\right)\cdot \exp\left(-\frac{x^{2}}{fb}\right)$$


The function 
ψ(x)
 captures the altered signal, incorporating both amplitude and phase information, comparable to the pitch and timing of a musical note. The term 
$\frac{1}{\sqrt{\pi \cdot fb}}$
 works as a volume control. The component 
exp(2πi·fc·x)
 functions as both a pitch shifter and time modifier, modulating 
ψ(x)
 with a complex exponential that modifies the center frequency and creates phase shifts. The term 
$\exp(-x^{2}/fb)$
 serves as a magnifying glass, adding a Gaussian window on 
ψ(x)
 that zooms in on a specified time interval around 
x=0
. The width of this window, determined by 
fb
, controls the zoom level and focuses emphasis on a particular instant in the signal. Together, these components constitute a Complex Morlet wavelet.

In signal processing and image analysis, the Mexican Hat wavelet function, also termed the Ricker wavelet or the second derivative of a Gaussian, is a mathematical function that is commonly exploited. The following equation defines this below (Equation 8):
(8)
$$\psi \left(t\right)=A\cdot \left(1-t^{2}\right)\cdot e^{-\frac{t^{2}}{2\sigma^{2}}}$$


In this instance, the wavelet’s width is determined by 
σ
 and *A* acts as a normalization constant. Because of its zero-crossings, this wavelet is commonly applied in signal processing applications like edge detection. The Shannon wavelet function is defined as (Equation 9):
(9)
ψ(t)=πt·sin(2πat)·cos(2πbt)


Here, 
πt
 is a linear term, and *a* and *b* control the frequencies of the sine and cosine components. This wavelet is associated with wavelet analysis and is known for its mathematical simplicity. It is used in applications where signals need to be analyzed in both frequency and time domains.

This function combines a Gaussian window with a complex sinusoidal wave, allowing it to effectively capture oscillatory patterns in the data. The parameters 
σ
 and 
$f_{0}$
 can be adjusted to match the characteristics of the data under analysis. Through the convolution process, the selected mother wavelet slides along the signal, calculating inner products at different positions via coefficient calculation. The resulting coefficients represent the alignment between the wavelet and the signal at each point and scale.

#### 3.2.2. Spectral Image Generation

Images naturally exhibit spatial features and patterns, which are often arranged hierarchically, and this aligns with the strengths of CNNs. Thus, the spectral images of the signal are generated to extract the potential features using the convolution process for activity recognition. In this step, the resulting coefficients from the convolution in CWT are used to construct the scalogram images to find the patterns using CNNs. Scalogram offers a comprehensive two-dimensional depiction of how a signal’s frequency content changes over time. The x-axis of the scalogram corresponds to time, the y-axis indicates the frequency, and the intensity or color at each point represents the magnitude of the wavelet coefficients. The intensity or color mapping in a scalogram plays a pivotal role in conveying information about the amplitude or power of wavelet coefficients at distinct time and frequency coordinates. Brighter colors or greater intensity often signify higher amplitudes or powers, offering a clear visual representation of the signal’s strength at specific time-frequency locations, which helps to expose the characteristics of different activities. At each point within the time-frequency plane, the scalogram acts as a graphical manifestation of the wavelet transform’s response for a particular frequency and time scale. Strong responses at specific coordinates indicate a pronounced presence of that frequency during the corresponding time interval, facilitating the identification of critical temporal events or shifts in frequency components. The scalogram allows for visualizing localized features in both time and frequency. This visualization helps in identifying time-localized frequency components and patterns in the signal that may not be apparent in the time or frequency domain alone.

### 3.3. Image Generation for Deep Learning Model

The data for HAR usually contain the X, Y, and Z sensor data, which represents movement behaviors by acceleration in three axes. This section describes the CWT-generated scalogram image formulation to address the motion features in a three-dimensional space. This paper applied a two-step process to prepare these images for the deep learning model. First, gray scaling, and secondly, these grayscale images were merged into a multi-channel representation, resembling an RGB format. Gray scaling is a common preprocessing step in image analysis and computer vision tasks. The conversion from a color image to a grayscale image involves reducing the image’s dimensionality by representing each pixel with a single intensity value, indicating its brightness or darkness. The most straightforward and commonly used method for gray scaling is to take a weighted sum of the red, green, and blue color channels. The equation [50] for this process is:
(10)
Y=0.299R+0.587G+0.114B


In Equation (Equation 10), Y represents the grayscale intensity, and R, G, and B are the red, green, and blue color values of the original pixel. The coefficients 0.299, 0.587, and 0.114 are weights assigned to each color channel. The choice of weights in the grayscale conversion equation is influenced by the varying sensitivity of the human eye’s photo-receptors to different wavelengths, with higher sensitivity to green (around 555 nm), moderate sensitivity to red (around 650 nm), and lower sensitivity to blue (around 475 nm).

Secondly, multi-channel activity images are generated by stacking the converted grayscale scalogram images of the X-, Y-, and Z-axis to produce the RGB images for activity recognition. The RGB activity image conveys the information of human motion in three-dimensional space. This processing step is crucial for feeding the data into deep learning models providing the model with a richer input. It enhances the ability to learn from different aspects of the activity simultaneously. This will help the model to capture complex patterns. Figure 3 illustrates a multi-channel representation of three-axis scalogram image generation.

Krizhevsky et al. [51], Ciresan et al. [52], and Howard et al. [53] (as well as many other researchers), demonstrated CNNs’ ability to learn rotation and shift-invariant features, suggesting they can adapt to variations in data presentation. This adaptability, along with the success of data augmentation techniques in image classification, strengthens the argument that consistent channel order swapping during training and testing likely will not significantly impact model performance if the order is swapped for both. Altering the axes and colors in the image generation process for deep learning models, such as exchanging the colors of the Y-axis and Z-axis, will not affect the model training and prediction results as long as the alteration is consistently applied to both the training and testing datasets. In the context of RGB images derived from three-frame tensor data, where each frame represents a layer and the data resolution is (x × y) × 3 (corresponding to R, G, B channels), deep learning models typically extract features from 2D representations of these frames. Whether the RGB channels are originally mapped as (R, G, B) or altered to (B, G, R), the model’s training process will adapt to recognize patterns based on the consistent configuration used. During testing, the model will apply these learned features from the altered RGB configuration in a consistent manner. Therefore, the performance metrics, such as accuracy, are expected to remain comparable because the model’s internal feature representations will align with the consistently altered input data. This consistency ensures that the recognition capabilities are preserved, regardless of the specific channel assignment. However, for the purpose of this paper, RGB as the X-, Y- and Z-axes are considered to be preprocessed data.

### 3.4. Training Models for Extracting Deep Features

This section demonstrates the training process of deep learning models for extracting features from activity images, utilizing generated images for both training the models and extracting the features. To accomplish this, a selection of CNN architectures, including CNN, AlexNet, ResNet50, ResNet101, and MobileNetV3, are employed. Additionally, for classification tasks, the softmax classifier is utilized.

Following the training phase, the feature extraction process involves several key steps common across these architectures. Firstly, the activity image is forwarded through the chosen deep learning architecture, where convolutional layers apply filters to capture low-level features like edges and shapes. Subsequently, pooling layers downsample the data, reducing their dimensionality while retaining essential information. Then, fully connected layers amalgamate the extracted features from previous layers, forming higher-level representations of the image. Finally, the output of the last fully connected layer, situated before the softmax layer in classification tasks, serves as the feature vector. This vector encapsulates the critical characteristics of the input image, rendering it suitable for various tasks, including classification or further analysis. Advancements in deep learning have propelled the development of powerful activity models, particularly using Convolutional Neural Networks (CNNs). These models, designed for tasks such as image classification and visual recognition, leverage intricate architectures consisting of convolutional layers, ReLU activation functions, pooling layers, and fully connected layers. The training process involves backpropagation and optimization algorithms, dynamically adjusting filter weights to minimize the loss function. Among notable architectures, AlexNet introduced groundbreaking features, including ReLU activation functions and dropout regularization, setting standards for subsequent models. ResNet addressed challenges associated with increasing network depth by introducing skip connections and thus revolutionized the training of deep neural networks. Additionally, MobileNetV3, optimized for mobile hardware, balances between efficiency and accuracy, making it ideal for resource-constrained devices. This section explores the evolution and contributions of these deep learning models in the context of activity model training. In the realm of deep learning for tasks like activity recognition using wearable sensor data, each neural network architecture typically has specific requirements for the shape of input data it can accept. For instance, popular models, such as AlexNet, and ResNet variants like ResNet-101, have standardized input shapes that align with their design specifications. AlexNet typically expects inputs sized at [227, 227, 3]; on the other hand, ResNet models commonly use [224, 224, 3]. These dimensions correspond to the width, height, and number of color channels (RGB) of the input images or data representations. Therefore, to effectively utilize these models for activity recognition tasks, it is crucial to preprocess the sensor data into the appropriate shape that matches the requirements of the chosen deep learning architecture. This approach ensures that the models can process the data correctly and leverage their designed features optimally, thereby enhancing the accuracy and performance of activity recognition systems in applications such as smart homes and healthcare monitoring.

CNN: A Convolutional Neural Network (CNN) [54] model is designed to extract the visual feature of activity images for HAR. It consist of convolutional layers, ReLU activation functions, pooling layers, and fully connected layers, enabling automatic feature learning from raw data. Convolutional layers extract local features, ReLU introduces nonlinearity, pooling layers reduce spatial dimensions, and fully connected layers make classification decisions. Through backpropagation and optimization algorithms during training, filter weights are adjusted to minimize the loss function, resulting in state-of-the-art performance across visual recognition applications. The CNN architecture typically involves multiple convolutional layers with configurations like Conv2D employing 32/64/128/128 filters and 3 × 3 kernels, followed by ReLU activation and MaxPooling2D layers for spatial downsampling. A dropout layer with a rate of 0.5 is incorporated to mitigate overfitting. The resulting flattened output represents abstract features extracted from input images, crucial for capturing hierarchical patterns. These features are then classified using various classifiers, showcasing the effectiveness of CNNs in image analysis and recognition tasks.

AlexNet: The AlexNet [51], introduced by Alex Krizhevsky et al., marked a significant advancement in deep learning by pioneering the use of Rectified Linear Unit (ReLU) activation functions and dropout regularization in CNNs. This architecture, designed for large-scale image datasets, comprises five convolutional layers with max-pooling, followed by three fully connected layers, incorporating dropout for model complexity control. With approximately 60 million parameters, the network features convolution sizes ranging from 11 × 11 in the first layer to 3 × 3 in subsequent layers. ReLU activation accelerates model convergence, while dropout mitigates overfitting. Data augmentation techniques, such as random flipping and cropping, enhance the diversity of the training dataset. Notably, overlapping pooling is employed to reduce spatial dimensions while preserving information. The meticulous design choices in AlexNet laid the foundation for subsequent advancements in deep learning architectures, establishing enduring standards. The feature extraction process in AlexNet begins with an input shape of [227, 227, 3]. The architecture starts with a Conv2D layer with 96 filters, employing an 11 × 11 kernel size and a stride of 4. Batch normalization and ReLU activation follow, with subsequent MaxPooling2D operations (pool size: 3 × 3, stride: 2) facilitating spatial downsampling. Two additional Conv2D layers with 256 filters and 5 × 5 kernel size, accompanied by ReLU activation, are followed by another MaxPooling2D layer. Three Conv2D layers with 384 filters, 3 × 3 kernel size, and ReLU activation create a hierarchy of features, culminating in an additional Conv2D layer with 256 filters and 3 × 3 kernel, followed by ReLU activation. Flattening the output into a one-dimensional vector condenses abstracted features learned by the convolutional layers, facilitating the representation of high-level patterns for classification models.

ResNet: After the groundbreaking success of AlexNet in the 2012 ImageNet competition, subsequent neural network architectures sought to further improve performance by increasing depth. However, this approach presented challenges, such as the vanishing/exploding gradient problem. The ResNet (Residual Network) [55] architecture tackled this issue by introducing skip connections, forming residual blocks that facilitate the learning of residual mappings. ResNet focuses on learning residual mappings related to identity mapping, ensuring smoother gradient propagation and faster training. Its architecture typically consists of two 
3×3
 convolutional layers with Batch Normalization and ReLU activation, along with a 
1×1
 convolutional layer for deeper networks. Variants like ResNet-50, ResNet-101, or ResNet-151 address the trade-off between model complexity and computational efficiency, capturing intricate features while mitigating challenges associated with deep networks. The incorporation of skip connections enables practitioners to select a ResNet variant that balances model depth and computational efficiency for optimal performance. In the feature extraction process of ResNet-50, input images are standardized to a shape of [224, 224, 3]. The ResNet-50 model, with its 50-layer depth and innovative use of residual learning, initializes weights from the "imagenet" dataset and freezes all layers during feature extraction to leverage pre-trained knowledge and prevent overfitting. Feature extraction involves passing input images through the stacked layers of ResNet-50, including residual blocks designed for learning hierarchical features. The residual connections facilitate the direct flow of information through the network, addressing challenges associated with training very deep networks. Similarly, the ResNet-101 model extends the ResNet-50 architecture by incorporating additional layers, resulting in a deeper neural network with 101 layers. The increased depth of ResNet-101 allows for a more intricate representation of hierarchical features. The abstract features extracted from the input images by ResNet-50 and ResNet-101 are then classified and detected using conventional classifiers. However, ResNet-151 or higher layer ResNet models are not tested due to the significant trade-offs in computation and memory requirements. Deeper networks require substantially more computational power to train, as each additional layer increases the data processing and calculations needed during training and inference. This results in longer training times and higher resource consumption. Additionally, deeper models demand more memory to store the increased number of weights and intermediate activations, which can quickly exceed the hardware capacity, especially on GPUs with limited VRAM. Furthermore, as we do not have much data, deeper networks typically need larger datasets to avoid overfitting, and if the available dataset is not sufficiently large, a deeper model like ResNet151 might not generalize well and could perform worse than shallower models. Hence, despite the theoretical potential for better accuracy, the practical constraints of computational power, memory, and dataset size led to the decision not to use ResNet-151 in this case. MobileNetV3: MobileNetV3 [56], introduced in 2019 as the latest iteration in the MobileNet series, signifies a significant breakthrough in mobile and embedded vision applications. Designed as an optimized version of EfficientNet for mobile hardware, MobileNetV3 addresses the critical challenge of balancing efficiency and accuracy in resource-constrained devices. Its architecture introduces innovative elements like the ’hardSwish’ activation function, an enhancement over ’Swish’. It incorporates techniques such as half-removed Squeeze and Excitation (SE) blocks alongside Neural Architecture Search (NAS) technology. These advancements empower MobileNetV3 to achieve superior results by strategically reducing network complexity, making it exceptionally well-suited for deployment on smartphones, tablets, and Internet of Things (IoT) devices where memory and power limitations are paramount. MobileNetV3’s flexibility is demonstrated through its robust performance across convolutional neural networks with varying layer counts, establishing it as a versatile solution compared to other architectures. In the feature extraction process of MobileNetV3, particularly utilizing the “small” variant, input images are standardized to a shape of [227, 227, 3]. The selection of MobileNetV3 small is based on its efficiency, leveraging depthwise separable convolution layers and linear bottleneck layers to create a lightweight and computationally efficient model adaptable to diverse applications. Additionally, the model incorporates features like a configurable width multiplier and L2 regularization, offering flexibility and enhanced generalization. The output tensor from MobileNetV3 small represents a concise set of abstract features extracted from the input images, leveraging depthwise separable convolutions for efficient feature learning.

### 3.5. Classification with Softmax Classifier

The softmax classifier is applied to retrieve feature data of the instances of HAR scalogram images in the proposed research.

Softmax, also known as Multinomial Logistic Regression, is a widely used method for multiclass classification problems. It extends binary logistic regression to handle multiple classes by employing the softmax function to transform raw predictions into probabilities. Given a set of input features *x* and a weight matrix *W* along with a bias vector *b*, the unnormalized log probabilities for each class *i* are calculated using the equation: 
$z_{i}=W_{i}\cdot x+b_{i}$
. Here, 
$W_{i}$
 represents the 
i−th
 row of the weight matrix *W*, and 
$b_{i}$
 is the 
i−th
 element of the bias vector *b*. The softmax function then converts these log probabilities into a valid probability distribution over all classes. For class *j*, the softmax function is defined as:
(11)
$$softmax{\left(z\right)}_{j}=\frac{e^{zj}}{\sum_{k=1}^{K}e^{zk}}$$

where *K* is the total number of classes. The final prediction is made by selecting the class with the highest probability. The training objective typically involves minimizing the cross-entropy loss, which measures the dissimilarity between the predicted probabilities and the true distribution of classes.

## 4. Experiment and Result Analysis

The following section of this paper outlines the process of generating scalogram images and detecting activity using deep learning models.

### 4.1. Dataset Description

The research employs wearable sensors, particularly accelerometers and gyroscopes, for the generation of activity data. In this study, the SisFall [57] and PAMAP2 [58] datasets are utilized, which consist of recordings from wearable sensors worn by the subjects during various physical activities. The aim is to extract meaningful information for activity recognition and monitoring tasks.

The SisFall dataset features two accelerometer sensors that produce acceleration data in three axes (X, Y, Z) and the second sensor, a gyroscope, generates rotation signals in three axes (X, Y, Z) for a single activity. Each file in the dataset comprises nine columns representing nine subjects, with the number of rows varying depending on the duration of the tests. The subjects generate the data by wearing waist-worn sensor devices. These sensors operate at a sampling frequency of 200 Hz, enabling detailed and accurate data capture. The dataset encompasses a comprehensive set of 19 Activities of Daily Living (ADL) performed by two age groups. Elderly people group was comprised of 15 individuals (8 males aged 60–71 and 7 females aged 62–75) whereas young adults’ group was comprised of 23 participants (both males and females aged 19–30). A list of these Activities of Daily Living (ADL) is provided in Table 1.

To further validate the proposed method, the PAMAP2 dataset is also used to evaluate its performance on a different set of real-world data. In the PAMAP2 dataset, 9 subjects participated in 18 activities while wearing IMU sensors and a heart rate monitor. An IMU (Inertial Measurement Unit) is a device that combines an accelerometer, gyroscope, and magnetometer to measure and report a body’s specific force, angular rate, and orientation. Each IMU is attached to the chest, dominant arm, and dominant-side ankle. The dataset includes 12 different protocol activities and 6 optional activities. Each participant performed the protocol activity and some of the optional activities. Each file in the dataset contains a subject’s data with 54 columns: one for timestamp, one for activity label, and the remaining 52 for raw sensory data. These 52 columns hold data from the 3 IMU sensors, each containing 17 columns. The IMUs record temperature, 13-bit 3-bit accelerometer data at a scale of 
A−16g
, 13-bit 3D accelerometer data at a scale of 
A−6g
, 3D gyroscope data, 3D magnetometer data, and orientation data, but these data are not valid for this dataset. The sampling frequency is around 100 Hz. The activities are shown in Table 2.

The activity names from the SisFall dataset have been shortened due to their detailed nature. In Table 3, the shortened forms of the activity details from the SisFall dataset are displayed. Within the scope of this work, an attempt is made to select activities of a similar nature. However, it is noteworthy that the SisFall dataset presents a more intricate array of activity combinations. Consequently, activities demonstrating superior outcomes are prioritized for initial inclusion. Detecting the remaining activities is reserved for future endeavors, where there is an aspiration to enhance activity detection effectiveness and precision.

For the dataset split, we randomly selected 70% for training, 20% for testing, and 10% for validation.

### 4.2. Image Generation and Preprocessing

A crucial step in extracting time-frequency representations of the signals involves converting the raw sensor data into scalogram images using CWT. To enhance the signal representation, a diverse set of mother wavelets, a broad range of scales, and various sampling frequencies are employed. This comprehensive approach aims to capture the intricate details and varied frequency components present in the signals. The selection of mother wavelets, including the Morlet wavelet, Complex Morlet wavelet, Shannon wavelet, and Mexican Hat wavelet, allows for a nuanced exploration of different signal characteristics.

The application of these mother wavelets spans across a range of scales from 1 to 128, facilitating a thorough analysis of the signal’s time-frequency features. Additionally, considering different sampling frequencies, precisely at 50, 100, and 200 Hz, adds further granularity to the investigation. This multi-faceted strategy achieves a more refined and comprehensive representation of the signals in the study on HAR. Among them, the best outcome, i.e., proper representation and accurate classification results, is obtained with the frequency of 200 Hz and the Morlet wavelet as the mother wavelet.

As illustrated in Figure 4, The preprocessing pipeline involved converting the raw sensor data into scalogram images and subsequently converting them into grayscale. Furthermore, the grayscale conversion is applied to the scalogram images by reducing the color space from three channels (RGB) to a single channel representing intensity. The subsequent merging of the grayscale images into RGB channels enables us to leverage the information from multiple sensor axes. We create a multi-channel representation of the data by representing each sensor axis as a separate channel. This approach provides the opportunity to capture and incorporate different aspects of the activity patterns into the analysis. Each channel represents a specific axis, enabling the exploration of independent information from different directions or orientations captured by the wearable sensors. Then the multi-channel representation of the scalogram images in RGB (R for the x-axis, G for the y-axis, and B for the z-axis) format facilitates the utilization of powerful deep learning models that operate on image data. Table 4 shows the processed images for different activities of the first accelerometer from the SisFall dataset with Shannon, Morlet, Complex Morlet and Mexican Hat mother wavelet successively. The choice of the mother wavelet function in wavelet transform analysis is crucial due to its significant impact on the accuracy and effectiveness of signal analysis tasks. Keng and Leong [59] highlight how different mother wavelets can lead to varying decomposition results, potentially compromising the accuracy of signal analysis tasks. Maneesh et. al. [60] further explore how the properties of mother wavelets influence their selection, affecting the detection of specific features within signals. Polikar’s [61] tutorial elucidates Heisenberg’s uncertainty principle, illustrating the trade-off between time and frequency resolution inherent in different mother wavelets. Ultimately, the convolution process between the signal and the chosen mother wavelet determines how well relevant features are extracted, with some wavelets better suited than others for specific tasks. This variability underscores why different wavelet transforms yield differing results in accuracy and effectiveness, emphasizing the critical role of the mother wavelet function in signal analysis.

### 4.3. HAR Feature Extraction Using Deep Learning

In the proposed research for Human Activity Recognition (HAR), various deep learning architectures including CNN, AlexNet, ResNet50, ResNet101, and MobileNetV3 are employed for feature extraction from individual images. Each architecture offers unique parameters and characteristics crucial for extracting meaningful features. For instance, the input image size typically adheres to a standard dimension, such as 
224×224
 pixels with three color channels (e.g., RGB), ensuring uniformity across models. CNNs, AlexNet, and ResNet architectures follow a series of convolutional layers, often followed by pooling layers, which reduce the spatial dimensions of the feature maps while retaining important information. AlexNet, being a deeper architecture, encompasses five convolutional layers and three fully connected (FC) layers. ResNet50 and ResNet101, characterized by their residual learning approach, typically comprise multiple residual blocks, with ResNet101 being deeper. These architectures culminate in fully connected layers, where the extracted features are aggregated to form a feature vector. MobileNetV3, optimized for efficiency, employs pointwise convolutions, width and resolution multipliers, and squeeze-and-excite blocks. Following feature extraction, a final classification is performed using a softmax classifier, often leading to activity recognition results. The models are trained using consistent hyperparameters, such as the Adam optimizer, categorical cross-entropy loss function, a learning rate of 0.01, 15 epochs, and a batch size of 3, to maintain simplicity and consistency across the models.

### 4.4. Activity Recognition

In this section, the features extracted from the previous stage are utilized for activity classification using a softmax classifier on the evaluation dataset. To assess the performance, several metrics, including Recall (R), Accuracy (A), F1-score (F1), and Precision (P), are employed to assess the proposed model. Recall (Equation (Equation 12)) quantifies the ability of a classification model to capture and correctly identify all relevant cases (positives) among the total actual positives.

(12)
$$R=\frac{TP}{TP+FN}\times 100$$


Accuracy (Equation (Equation 13)) represents the percentage of correct predictions out of the total predictions made by the model, providing an overall assessment of its performance.

(13)
$$A=\frac{TP+TN}{FP+TP+FN+TN}\times 100$$


F1-score (Equation (Equation 14)) serves as a metric that balances precision and recall, calculated using their harmonic mean.

(14)
$$F1=\frac{2\times (R\times P)}{R+P}\times 100$$


Precision (Equation (Equation 15)) is a metric assessing the accuracy of positive predictions, defined as the ratio of true positives to the sum of true positives and false positives.

(15)
$$P=\frac{TP}{TP+FP}\times 100$$


These equations quantify the performance metrics used to evaluate the proposed model’s effectiveness in activity detection. After feature extraction, the extracted features from CNN, AlexNet, ResNet-50, and MobileNetV3 are employed as inputs for the softmax classifier. Each model is compiled using appropriate loss functions, such as categorical cross-entropy, and optimizers, like Adam, to facilitate training for the classification task. Additionally, data augmentation parameters, including rescaling, rotation range of 40 degrees, width, and height shifting range of 0.2, shear range of 0.2, zoom range of 0.2, and horizontal flipping, are incorporated into the training pipeline to further improve model generalization and resilience to variations in input data.

Altogether, five popular architectures: simple CNN, ResNet50, ResNet101, MobileNetV3, and AlexNet are tested. Table 5 displays the overall performance of different models with different classifiers. Among them, it is found that the ResNet101 produces the best result with the Morlet wavelet and a scale range from 0 to 128 with SVM. This combination yields a classification precision of 98.2%, a recall of 97.0%, an F1-score of 97.6%, and an accuracy of 98.4% on the SisFall dataset. Notably, ResNet101 exhibits superior performance. The CNN achieves its maximum accuracy of 95.1% when utilized with the Mexican Hat wavelet. Similarly, AlexNet demonstrates improved performance with the Shannon wavelet, achieving an accuracy of 93.2%. Moreover, both ResNet50 and MobileNetV3 attain the highest accuracy when employed with the Morlet wavelet, achieving 94.9% and 95.8%, respectively. Figure 5 shows the confusion matrix of the SisFall dataset for the best combination, which is the ResNet101 and Morlet wavelet, providing a visual representation of the classification performance. In Table 6, the detailed classification report of the SisFall dataset is presented, including Precision, Recall, F1-score, and Accuracy for each class.

After achieving promising results on the SisFall dataset, the next phase of the study involved testing the same model architecture on the PAMAP2 dataset. This transition allowed us to explore the model’s adaptability and generalization capabilities across distinct datasets and activity recognition scenarios. The following section provides a detailed analysis of the attained results, highlighting the classification accuracy and other evaluations for each model architecture. Table 7 showcases all of the models evaluation metrics for each classifier. This dataset also yields optimal performance for ResNet101, achieving an Accuracy of 98.1%, an F1-score of 96.2%, a Precision of 98.1%, and a Recall of 95.3%. In contrast, among other models, CNN achieves the highest accuracy of 95.1% when paired with the Mexican Hat wavelet, while AlexNet demonstrates peak performance with an accuracy of 93.2% using the Shannon wavelet. Furthermore, ResNet50, coupled with the Morlet wavelet, achieves an accuracy of 94.9%, and MobileNetV3, under the same wavelet configuration, attains an accuracy of 95.8%.

Figure 6 displays the confusion matrix showcasing the excellent results obtained by the employed method. In Table 8, the detailed classification report of the PAMAP2 dataset is presented.

## 5. Comparison

In the comparative study presented in Table 9, the performance of the model was assessed using the PAMAP2 dataset and compared with several existing papers. Initially, the results were compared with those from Imen et al. [38], where an accuracy of only 54.4% was achieved using CWT-CNN 2D, and the best result of 0.622 was obtained with Random Forest. Another relevant study, Zeng et al. [62] reported an accuracy of 89.9% using LSTM with Continuous Temporal attention. Complexity issues were addressed by Xi et al. [63] through the utilization of Deep Dilated Convolutional networks, resulting in an accuracy of 93.2%. Qian et al. [64] utilized Distribution-Embedded Deep NN (DDNN) and an accuracy of 93.40% was achieved. Intriguingly, an image representation approach with CNN was employed by Sanchez et al. [65], and a remarkable breakthrough was achieved with an accuracy of 0.967 for repetitive movements, although slightly lower at 0.887 for posture recognition. In the conducted experiments, a maximum accuracy of 98.1% on the PAMAP2 dataset was obtained using a combination of Continuous Wavelet Transform (CWT) and ResNet-101, which represents the state-of-the-art accuracy achieved so far in image representation for the PAMAP2 dataset.

Table 10, displays the existing papers’ results alongside the proposed model’s result. Notably, there has been limited prior research on activity detection using image representation in the SisFall dataset. In a study conducted by Abbas et al. [66], the authors utilized a 4-2-1 1D spatial pyramid pooling approach with Haar wavelet features and employed various classifiers, including K-Nearest Neighbors (KNN), Support Vector Machines (SVM), Random Forests, and extreme Gradient Boosting (XGB). Their achievements included an F1-score of 94.67% for fall and activity detection. However, it is important to highlight that while their model was trained on the SisFall dataset, the evaluation was conducted using their proprietary dataset. On the other hand, Al-majdi et al. [67], adopted a CNN-LSTM-based method and achieved a maximum accuracy of 94.18%. In this proposed method, a maximum accuracy of 98.4% was attained in Resnet-101 for activity detection, representing an almost 3% improvement compared to existing work.

This research demonstrates the efficacy of using scalogram images generated from wearable sensor data for human activity recognition through deep learning models. We compare the performance with the 1D sensor-based approach, and the proposed model outperforms other state-of-the-art models. In addition, we use different deep learning models to evaluate and compare their performance. However, in yielding this performance, we maintained consistent default model parameters to ensure fairness in comparison as this study does not focus on optimizing models for maximum performance.

## 6. Conclusions

In this study, an innovative Human Activity Recognition (HAR) model is introduced, which transforms raw accelerometer signal data into informative scalogram images using the Continuous Wavelet Transform (CWT) with an appropriate mother wavelet function. By converting these images to grayscale and merging data across three axes, this approach captures discriminant temporal and frequency patterns crucial for activity recognition. The visual interpretability provided by scalogram images, combined with the subsequent integration of deep learning models for feature extraction, yields a deep feature set for activity recognition. The proposed feature extraction model, coupled with classifier, offers an efficient HAR solution. This method’s ability to convert wearable sensor data into visually interpretable scalogram images underscores its efficacy in human activity recognition. The proposed approach is validated using two benchmark datasets, and the results demonstrate robust performance in classification, surpassing existing state-of-the-art methods, yielding 98.4% and 98.1% for the SisFall and PAMAP2 datasets, respectively. Overall, this approach represents a significant advancement in the field of HAR, offering a versatile and accurate solution with potential implications for a range of practical applications, including healthcare monitoring. The proposed model not only outperforms other state-of-the-art models but also provides a basis for future innovations in this domain.

## Figures and Tables

**Figure 1 sensors-24-04343-f001:**
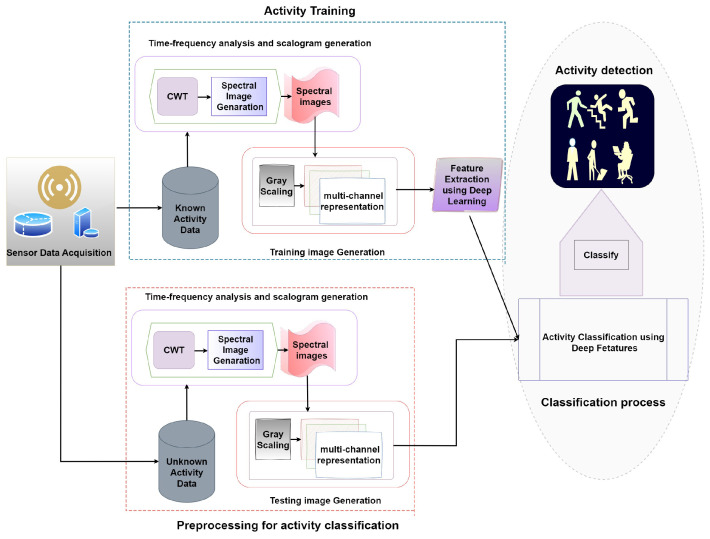
Flowchart of the proposed human activity recognition methodology.

**Figure 2 sensors-24-04343-f002:**
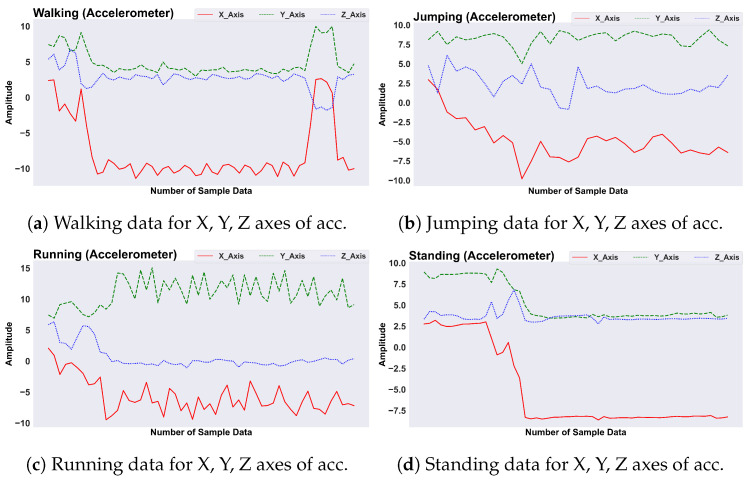
Sensor-generated signals for the X, Y, and Z axes from from accelerometer for different activities.

**Figure 3 sensors-24-04343-f003:**
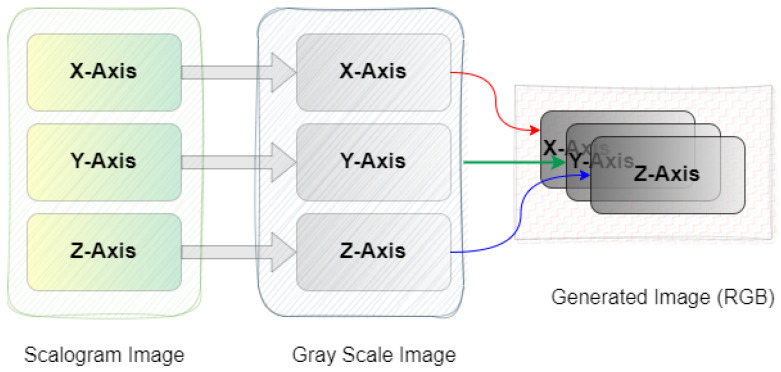
Process of RGB image generation.

**Figure 4 sensors-24-04343-f004:**
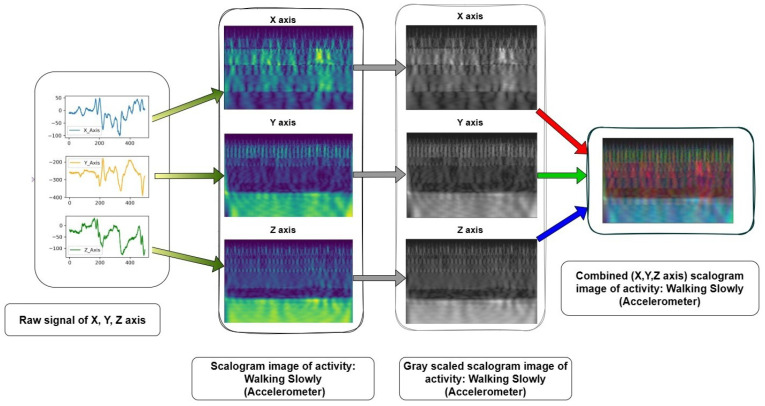
An example of activity image generation from sensor data.

**Figure 5 sensors-24-04343-f005:**
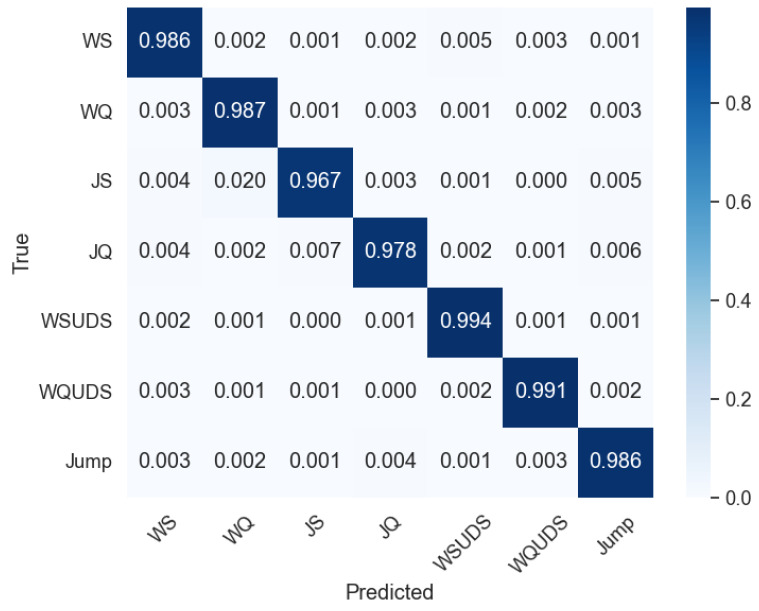
Confusion matrix of activity classification using ResNet-101 and Morlet wavelet with SisFall dataset.

**Figure 6 sensors-24-04343-f006:**
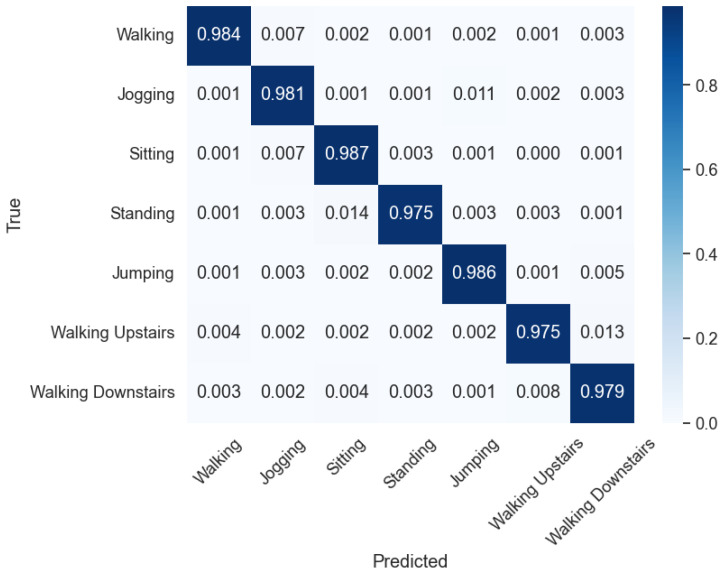
Confusion matrix of activity classification using ResNet-101 and Morlet wavelet with PAMAP2 dataset.

**Table 1 sensors-24-04343-t001:** Activity of SisFall dataset.

I Code	Activity
01	Walking slowly
02	Walking quickly
03	Jogging slowly
04	Jogging quickly
05	Walking upstairs and downstairs slowly
06	Walking upstairs and downstairs quickly
07	Slowly sit in a half-height chair, wait a moment, and up slowly
08	Quickly sit in a half-height chair, wait a moment, and up quickly
09	Slowly sit in a low height chair, wait a moment, and up slowly
10	Quickly sit in a low height chair, wait a moment, and up quickly
11	Sitting a moment, trying to get up, and collapse into a chair
12	Sitting a moment, lying slowly, wait a moment, and sit again
13	Sitting a moment, lying quickly, wait a moment, and sit again
14	Being on one’s back, change to the lateral position, wait a moment, and change to one’s back
15	Standing, slowly bending at knees, and getting up
16	Standing, slowly bending without bending knees, and getting up
17	Standing, get into a car, remain seated and get out of the car
18	Stumble while walking
19	Gently jump without falling (trying to reach a high object)

**Table 2 sensors-24-04343-t002:** Activity of PAMAP2 dataset.

ActivityID	Activity	ActivityID	Activity
01	Lying	11	Car driving
03	Standing	13	Descending stairs
04	Walking	16	Vacuum cleaning
05	Running	17	Ironing
06	Cycling	18	Folding Laundry
07	Nordic Walking	19	House Cleaning
09	Watching TV	20	Playing soccer
10	Computer work	24	Rope Jumping
		0	Other (Transient activity)

**Table 3 sensors-24-04343-t003:** Short forms of activities of SisFall dataset.

Activity	Short Form
Walking slowly	WS
Walking quickly	WQ
Jogging slowly	JS
Jogging quickly	JQ
Walking upstairs and downstairs slowly	WSUDS
Walking upstairs and downstairs quickly	WQUDS
Gentle jump without falling	Jump

**Table 4 sensors-24-04343-t004:** Generated scalogram images of some activities.

Activity|Wavelets	Shannon	Morlet	Complex Morlet	Mexican Hat
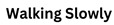	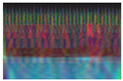	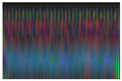	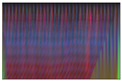	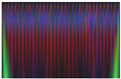
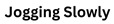	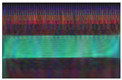	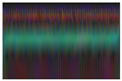	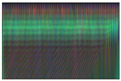	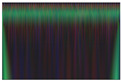
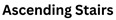				
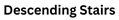			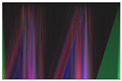	
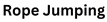	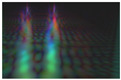	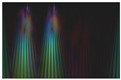	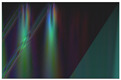	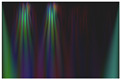

**Table 5 sensors-24-04343-t005:** The overall performance of different DL models with different mother wavelets in SisFall.

Model	Mother Wavelet	Precision	Recall	F1-Score	Accuracy (%)
CNN	Shannon	0.920	0.900	0.910	91.2
	Morlet	0.942	0.948	0.945	94.5
	Complex Morlet	0.910	0.890	0.900	90.2
	**Mexican Hat**	**0.955**	**0.945**	**0.950**	**95.1**
AlexNet	**Shannon**	**0.929**	**0.925**	**0.922**	**93.2**
	Morlet	0.905	0.895	0.900	90.3
	Complex Morlet	0.870	0.850	0.860	86.3
	Mexican Hat	0.880	0.860	0.870	87.1
ResNet50	Shannon	0.910	0.900	0.910	90.8
	**Morlet**	**0.951**	**0.947**	**0.949**	**94.9**
	Complex Morlet	0.890	0.880	0.880	88.9
	**Mexican Hat**	0.870	0.850	0.860	86.3
ResNet101	Shannon	0.97	0.96	0.96	96.8
	**Morlet**	**0.982**	**0.970**	**0.976**	**98.4**
	Complex Morlet	0.940	0.950	0.950	94.9
	Mexican Hat	0.925	0.915	0.920	92.1
MobileNetV3	Shannon	0.930	0.910	0.920	92.5
	**Morlet**	**0.960**	**0.955**	**0.958**	**95.8**
	Complex Morlet	0.940	0.930	0.930	93.7
	Mexican Hat	0.895	0.885	0.890	89.2

**Table 6 sensors-24-04343-t006:** Precision, Recall, F1-score, and Accuracy for each class of SisFall.

Class	Precision	Recall	F1-Score	Accuracy (%)
WS	0.988	0.984	0.985	98.6
WQ	0.975	0.977	0.976	98.7
JS	0.958	0.960	0.959	96.8
JQ	0.964	0.965	0.964	97.8
WSUDS	0.992	0.991	0.992	99.4
WQUDS	0.988	0.990	0.989	99.1
Jump	0.991	0.992	0.992	98.6

**Table 7 sensors-24-04343-t007:** The overall performance of different DL models with different mother wavelets in PAMAP2.

Model	Mother Wavelet	Precision	Recall	F1-Score	Accuracy %
CNN	Shannon	0.932	0.915	0.922	91.8
	**Morlet**	**0.947**	**0.924**	**0.935**	**96.5**
	Complex Morlet	0.952	0.936	0.943	95.8
	Mexican Hat	0.947	0.939	0.953	94.5
AlexNet	**Shannon**	**0.965**	**0.943**	**0.953**	**97.2**
	Morlet	0.958	0.946	0.952	95.2
	Complex Morlet	0.939	0.912	0.924	94.5
	Mexican Hat	0.945	0.922	0.933	94.2
ResNet50	Shannon	0.957	0.935	0.943	94.8
	**Morlet**	**0.976**	**0.961**	**0.958**	**97.8**
	Complex Morlet	0.964	0.948	0.953	96.7
	Mexican Hat	0.959	0.943	0.951	95.1
ResNet101	Shannon	0.974	0.952	0.963	95.2
	**Morlet**	**0.981**	**0.953**	**0.962**	**98.1**
	Complex Morlet	0.973	0.947	0.958	96.3
	Mexican Hat	0.944	0.91	0.933	92.8
MobileNetV3	**Shannon**	**0.956**	**0.939**	**0.948**	**97.4**
	Morlet	0.950	0.933	0.941	94.7
	Complex Morlet	0.946	0.929	0.937	92.5
	Mexican Hat	0.943	0.918	0.929	94.1

**Table 8 sensors-24-04343-t008:** Precision, Recall, F1-score, and Accuracy for each class of PAMAP2.

Class	Precision	Recall	F1-Score	Accuracy
Walking	0.982	0.984	0.983	0.984
Jogging	0.981	0.983	0.982	0.981
Sitting	0.988	0.985	0.987	0.987
Standing	0.974	0.976	0.975	0.975
Jumping	0.987	0.987	0.987	0.986
Going Upstairs	0.976	0.975	0.976	0.975
Going Downstairs	0.979	0.978	0.978	0.979

**Table 9 sensors-24-04343-t009:** Comparison of results on PAMAP2 dataset.

Study	Method	Accuracy (%)
Imen et al. [38]	CWT-CNN 2D	54.4
Imen et al. [38]	Random Forest	62.2
Zeng et al. [62]	LSTM with Continuous Temporal Attention	89.9
Xi et al. [63]	Deep Dilated Convolutional Networks	93.2
Qian et al. [64]	Distribution-Embedded Deep NN (DDNN)	93.40
Sanchez et al. [65]	CNN for Image Representation (Repetitive)	96.7
Sanchez et al. [65]	CNN for Image Representation (Posture)	88.7
**Proposed Method**	CWT and ResNet-101	98.1

**Table 10 sensors-24-04343-t010:** Comparison of results on SisFall dataset.

Study	Method	Accuracy (%)
Abbas et al. [66]	4-2-1 1D Spatial Pyramid + Haar Wavelet	94.67
Abbas et al. [66]	KNN, SVM, Random Forest, XGB (Proprietary)	Not Specified
Al-majdi et al. [67]	CNN-LSTM-Based Method	94.18
**Proposed Method**	CWT and ResNet-101	98.4

## Data Availability

No new data were created or analyzed in this study. Data sharing is not applicable to this article.

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
