# Peer review of "A Robust Deep Feature Extraction Method for Human Activity Recognition Using a Wavelet Based Spectral Visualisation Technique"

_sensors, 2024, doi:10.3390/s24134343_

Round 1

Reviewer 1 Report

Comments and Suggestions for Authors

Accurate recognitions of human activities have great potential for future applications in smart homes, intelligent monitoring, and related fields. This paper integrates deep learning models and continuous wavelet transforms to extract detailed features of human activities, which contribute to the advancing in the technology of Human Activity Recognition (HAR). The convincing experimental results and analyses presented surpass existing methods, making contributions to the field.

While, before considering for publication in the journal of Sensors, I suggest the authors to answer the following comments and make revisions.

1.       In Section 3.1 of this paper, the authors mentioned that incorporating additional gyroscope data into the model did not effectively improve HAR. Generally, it is expected that more input information would lead to better estimations. I suggest the authors to give a tentative explanation for this.

2.       In Figure 2, it appears that Figures 2a and 2d are identical. It is unclear whether this is an error? If the accelerometer readouts for the two movement patterns are essentially the same, it is advisable to clarify this in this paper.

3.       For the image generation with deep learning models, R corresponds to the X-axis, G corresponds to the Y-axis, and B corresponds to the Z-axis. Would altering the axes and colors (such as exchange the colors of the Y-axis and Z-axis) affect the model training and prediction results?

4.       For different deep learning models, is there a standardization of input data shapes, such as [227, 227, 3] and [224, 224, 3], or are there distinct data preprocessing pipelines tailored to the specific requirements of each model?

5.       In Table 5. Under the ResNet50 model, there are two entries for the mother wavelet, both labeled as “Complex Morlet”.

6.       According to the results, it appears that the ResNet101 model outperforms the ResNet50 model. Have there been any attempts to test deeper network architectures such as ResNet151?

7.       In Table 9 and Table 10, the references do not match the authors names listed in the tables. It is advisable for the authors to ensure the consistency.

8.       We suggest the authors to briefly discuss in the Comparison or Conclusion section the reasons for the differences in the results when using different wavelet transforms.

Author Response

Dear reviewer

We would like to thank you for your precious time reviewing our paper and providing us with your valuable comments. The comments were immensely helpful in improving the quality of our manuscript. We have revised the manuscript and have carefully addressed all the comments. The corresponding changes and refinements we made in the revised manuscript are summarized in our response report.

Please find the attachment and have a look.

Thank you

Sincerely,

Islam et al.

Reviewer 2 Report

Comments and Suggestions for Authors

This paper presents a method that recognizes different daily life activities using time-frequency features of IMU sensors. Different wavelet transformation techniques were implemented and tested to find the best performance with the public datasets. There are several comments that should be addressed to improve the quality of the manuscript.

1.      This reviewer suggests considering moving the contributions to the conclusion section because the current presence of the contributions sounds less convincing. In addition, the third and fourth contributions are not the original contributions to the body of knowledge. These are just some tasks done in this study.

2.      Recent sensor-based deep learning models could be added to the literature review section because the current section describes relatively traditional methods.

3.      Lines 235 to 237 are confusing. Does it mean that the accelerometer generates its sensor values for only a single activity? Both the first sensor (accelerometer) and the second sensor (gyroscope) should generate signals in three axes for each activity. The conjunction “In contrast” makes it even more confusing.

4.      Lines 239 to 247 explain the rationale for utilizing accelerometer data in generating a scalogram in this study. The examples in Figure 2 show that the accelerometer data can represent the activities effectively. However, what about other activities like twisting the upper body? If this paper focuses on some particular activities for healthcare purposes, relevant explanations for selecting the type of sensors and data are highly expected.

5.      It is recommended to provide a more convincing rationale for using time-frequency domain values rather than a sequence of raw data. Also, as spatial-temporal features can be considered in other ways with raw sensor data, more explanations of the benefits of using the time-frequency domain value are recommended.

6.      Having consistency across the models is a clearly simple way. However, this reviewer is not sure whether this setting yields the best performance from each model. It may be true that the models have enough variables to make them distinguishable. But, a more detailed explanation is expected regarding this matter.

Comments on the Quality of English Language

there are several grammar issues in the manuscript.

Author Response

(The authors gave the same response as above.)

Round 2

Reviewer 2 Report

Comments and Suggestions for Authors

The comments have been addressed.